# Integrated Analysis of Genomic and Genome-Wide Association Studies Identified Candidate Genes for Nutrigenetic Studies in Flavonoids and Vascular Health: Path to Precision Nutrition for (Poly)phenols

**DOI:** 10.3390/nu16091362

**Published:** 2024-04-30

**Authors:** Tatjana Ruskovska, Filip Postolov, Dragan Milenkovic

**Affiliations:** 1Faculty of Medical Sciences, Goce Delcev University, 2000 Stip, North Macedonia; tatjana.ruskovska@ugd.edu.mk (T.R.); filip.153317@student.ugd.edu.mk (F.P.); 2Department of Nutrition, University of California, Davis, Davis, CA 95616, USA

**Keywords:** polyphenols, interindividual variability, genetic polymorphisms, hypertension, atherosclerosis, arterial stiffness, cardiovascular, nutrigenomics, nutrigenetics

## Abstract

Flavonoids exert vasculoprotective effects in humans, but interindividual variability in their action has also been reported. This study aims to identify genes that are associated with vascular health effects of flavonoids and whose polymorphisms could explain interindividual variability in response to their intake. Applying the predetermined literature search criteria, we identified five human intervention studies reporting positive effects of flavonoids on vascular function together with global genomic changes analyzed using microarray methods. Genes involved in vascular dysfunction were identified from genome-wide association studies (GWAS). By extracting data from the eligible human intervention studies, we obtained 5807 differentially expressed genes (DEGs). The number of identified upstream regulators (URs) varied across the studies, from 227 to 1407. The search of the GWAS Catalog revealed 493 genes associated with vascular dysfunction. An integrative analysis of transcriptomic data with GWAS genes identified 106 *candidate DEGs* and 42 *candidate URs*, while subsequent functional analyses and a search of the literature identified 20 top priority candidate genes: *ALDH2*, *APOE*, *CAPZA1*, *CYP11B2*, *GNA13*, *IL6*, *IRF5*, *LDLR*, *LPL*, *LSP1*, *MKNK1*, *MMP3*, *MTHFR*, *MYO6*, *NCR3*, *PPARG*, *SARM1*, *TCF20*, *TCF7L2*, and *TNF*. In conclusion, this integrated analysis identifies important genes to design future nutrigenetic studies for development of precision nutrition for polyphenols.

## 1. Introduction

(Poly)phenols are the most abundant bioactive compounds of plant origin and are present in the human diet in relatively large amounts, ranging from less than 500 to more than 1500 mg/d [1,2]. In the US adult population, a dietary intake of approximately 900 mg per 1000 kcal/d has recently been reported [3]. (Poly)phenols present extraordinary heterogeneity in their chemical structures, with over 500 chemical entities identified in the human diet, which are divided into two classes: flavonoids and non-flavonoids [4,5]. Following intake of (poly)phenols, they are metabolized by both human enzymes and gastro-intestinal microbiota. Once in the gut and intestine, (poly)phenols can be absorbed by enterocytes, enter the liver, and be converted by Phase I (oxidation, hydrolysis, and reduction) and Phase II (glucuronidation, methylation, and sulfation) metabolic reactions. Derived metabolites enter general circulation and can reach tissues. Non-absorbed (poly)phenols enter the colon where they can be directly metabolized by gut microbiota and give rise to low-molecular-weight metabolites. These gut microbiota-derived metabolites can be absorbed by the enterocytes and be further metabolized by Phase I and Phase II metabolic reactions in the liver before entering general circulation [6].

Flavonoids are among the best studied plant food bioactives in terms of their health-promoting properties. Human studies have shown that a diet rich in flavonoids can reduce type 2 diabetes risk [7], improve insulin sensitivity and blood lipids [8], and have beneficial effects on vascular function [9]. Recently, a long-term, large-scale, randomized, double-blind, placebo-controlled trial with cocoa extract supplementation [500 mg flavanols/d, including 80 mg (−)-epicatechin] showed a significant reduction in cardiovascular disease death by 27% among older adults [10]. In addition, molecular mechanisms underlying vasculoprotective effects of flavonoids have been investigated using omics technologies, such as transcriptomics, epigenomics, proteomics, and metabolomics [11]. These state-of-the-art untargeted analytical methods enable the identification of global molecular modulations, while subsequent bioinformatic analyses of individual omics data indicate key cellular pathways and regulatory mechanisms involved.

Despite the general trend demonstrating positive effects of dietary flavonoids on vascular health in humans, some less convincing results have also been reported in the literature. Such data have allowed for the identification of subgroups of participants where the vasculoprotective effects of flavonoids were more pronounced [12]. Other studies identified factors underlying the interindividual variabilities in health-promoting effects of flavonoids that include sex, age, ethnicity, body mass index, health status, gut microbiome, and genetic factors [13]. Of these, genetic factors have been the least studied.

Vascular function, which is one of the key determinants of overall health, is largely influenced by age [14] and lifestyle [15,16], with genetic factors also playing a significant role. A clinical study has shown the association of endothelial nitric oxide synthase (Nitric Oxide Synthase 3, *NOS3*) gene G894T polymorphism with hypertension risk and complications [17]. Also, monocyte chemoattractant protein-1 (*MCP-1*; or C-C Motif Chemokine Ligand 2, *CCL2*) gene −2578A > G polymorphism has been associated with an increased risk of coronary atherosclerosis in an asymptomatic population [18]. On the other hand, studies on flavonoids, genetic polymorphisms, and vascular health are scarce. The most convincing results were obtained from the studies that were focused on polymorphisms of well-established genes that are directly associated with vascular function or genes involved in the metabolism of circulating lipoproteins. It has been shown that the Glu298Asp single nucleotide polymorphism (SNP) in the *NOS3* gene differentially affects the vascular response to acute consumption of fruit and vegetables [19], and that polymorphisms in Apolipoprotein A1 (*APOA1*_rs964184) and Lipoprotein Lipase (*LPL*_rs12678919) genes determine the vasculoprotective effects of orange juice [20].

Given the small number of identified genetic polymorphisms that determine the interindividual variabilities in the effects of dietary flavonoids on vascular function, there is a need for additional studies focused on: (a) identification of candidate genes on dietary flavonoids and vascular function, and (b) testing of identified candidate genes in appropriately designed nutrigenetic studies. To address the first goal, i.e., identification of candidate genes for future nutrigenetic studies on dietary flavonoids and vascular function, one approach is through the integration of (i) data from human intervention studies with flavonoids showing modulations in global gene expression together with positive effects on vascular function and (ii) results from genome-wide association studies (GWAS) that have identified variants and risk alleles associated with vascular dysfunction, such as hypertension, atherosclerosis, or arterial stiffness. So far, such an innovative and powerful approach combining genomic and GWAS datasets has enabled the identification of candidate genes associated with, and directly governing, disease pathobiology [21,22], thus facilitating targeted studies to identify functional impact of major causal genes. Indeed, GWAS can identify hundreds of candidate genes associated with the development of disease, revealing a need for a systematic way to understand the causal mechanism(s) of these genes and a means to prioritize them for further study. The integration of genomic and GWAS data has recently allowed to identify candidate genes causal of coronary artery disease (CAD) [23]. The authors performed a comprehensive integrative analysis by combining CAD genome-wide association studies datasets (UK Biobank and CARDIoGRAMplusC4D) with transcriptomic data from the STARNET study (Stockholm–Tartu Atherosclerosis Reverse Network Engineering Task). The same approach has also been recently used to identify genes associated with atrial fibrillation (AF) [24]. The authors concluded that such an integrative omics strategy has improved the power of identifying AF-related genes compared to using GWAS alone.

Therefore, the aim of this study was to identify candidate genes whose polymorphisms potentially determine the interindividual variabilities of the effects of flavonoids on vascular health. To this end, we conducted integrative and functional analyses of genomic data from human intervention studies, presenting vasculoprotective effects of flavonoids and data from GWAS related to vascular dysfunction. Such a novel approach allowed us to identify top-priority candidate genes for future nutrigenetic studies on flavonoids and interindividual variability in vascular health effects, studies that will provide central leads for the development of precision nutrition for (poly)phenols.

## 2. Materials and Methods

This study is based on our previous systematic literature search and analysis of nutrigenomic effects of (poly)phenols related to cardiometabolic health in humans [25]. However, here we have only included the studies that demonstrated positive effects of flavonoids on vascular function and analyzed their genomic effects using microarray methods [26,27,28]. In addition, two recent studies of relevance for our analyses [29,30] were also included. In all of these studies, the analyses of global gene expression were conducted in samples of peripheral blood.

The analyses of upstream regulators (URs) of each set of differentially expressed genes (DEGs) were conducted using QIAGEN Ingenuity Pathway Analysis (IPA) on-line bioinformatic tool (https://digitalinsights.qiagen.com/, accessed on 28 July 2022, 29 July 2022, 5 August 2022, and 12 September 2022). For identification of genes that are associated with vascular dysfunction across published genome-wide association studies, we searched, in July 2022, GWAS Catalog (https://www.ebi.ac.uk/gwas/home, accessed on 16 July 2022) [31]. Pathway enrichment analyses were conducted using GeneTrail3.2 (https://genetrail.bioinf.uni-sb.de/, accessed on 19 July 2022) [32] as a platform to access the following two databases: Kyoto Encyclopedia of Genes and Genomes (KEGG) [33] and WikiPathways [34,35]. InteractiVenn (http://www.interactivenn.net/, accessed on 23 July 2022) [36] was used as a tool to retrieve elements that different datasets have in common. To identify variants with the highest frequencies of selected top-priority candidate genes, we interrogated the Variation Viewer database, National Center for Biotechnology Information, National Library of Medicine, National Institutes of Health, U.S. (https://www.ncbi.nlm.nih.gov/variation/view, accessed on 12 September 2022). Pharmacology-relevant variants of selected top-priority candidate genes were identified using the PharmGKB database (https://www.pharmgkb.org, accessed on 12 September 2022) [37,38].

GWAS-reported variants of selected top-priority candidate genes associated with vascular dysfunction were identified back in the GWAS Catalog (https://www.ebi.ac.uk/gwas/home, accessed on 16 July 2022) [31], while their frequencies in the global population and previously reported clinical significance were identified in the dbSNP database, National Center for Biotechnology Information, National Library of Medicine, National Institutes of Health, U.S. (https://www.ncbi.nlm.nih.gov/snp, accessed on 23 April 2023).

Gene names and symbols were searched in GeneCards database (https://www.genecards.org, accessed on 4 August 2022) [39]. The names of canonical pathways are presented as they appear in the interrogated databases.

A flowchart of the study is presented in Figure 1.

## 3. Results

### 3.1. Flavonoids Affect Global Gene Expression in Human Peripheral Blood Cells

#### 3.1.1. General Overview of Selected Studies and DEGs

Based on our previously reported strategy for the systematic literature search [25], we identified five human intervention studies with flavonoids that analyzed global gene expression in peripheral blood cells and demonstrated at least one positive effect on vascular function. General information about each of these studies, referred to as Paper 1 to Paper 5 (Paper 1 [26]; Paper 2 [27]; Paper 3 [28]; Paper 4 [30]; Paper 5 [29]) is presented in Table 1. In these studies, flavonoids of different subclasses were studied: flavanones in Paper 1 and Paper 5, flavanols in Paper 2 and Paper 4, or anthocyanins in Paper 3. Study populations differed across the selected studies and included overweight men (Paper 1), non-obese healthy male smokers (Paper 2), healthy men (Paper 3 and Paper 4), or postmenopausal women (Paper 5). For each of these studies, at least one positive effect on vascular function was reported, in the same (as for Paper 3) or in an associated paper (Morand et al. [40] for Paper 1; Weseler et al. [41] for Paper 2; Sansone et al. [42] for Paper 4; Habauzit et al. [43] for Paper 5) (Table 1).

The number of DEGs varies across the studies: *n* = 1693; 717; 554; 2231; and 1401 for Papers 1; 2; 3; 4; and 5 respectively. After removing duplicates, the total number of flavonoid-modulated genes reached *n* = 5807 (Appendix A). Comparative analysis of DEGs across the selected studies showed that *n* = 720 genes were in common for at least two studies, *n* = 67 genes were in common for at least three studies, and only two genes were in common for four studies. There were no DEGs that all five studies had in common (Figure 2; Appendix A).

#### 3.1.2. Upstream Regulators of DEGs

When analyzing the modulations in global gene expression, for biological interpretation of obtained experimental data, it is of particular importance to predict the upstream regulators (URs) of DEGs. For this analysis, we used the Qiagen IPA on-line bioinformatic tool (https://digitalinsights.qiagen.com/, accessed on 28 July 2022, 29 July 2022, 5 August 2022, and 12 September 2022), applying the default settings suggested by the manufacturer. With this analysis, for each set of DEGs, i.e., for each paper separately, we obtained URs that include not only protein coding genes and miRNAs, but also different chemical compounds, drugs, toxins, etc. The number of identified URs varied across the studies, from 227 to 1407, i.e., *n* = 227; 503; 508; 1407, and 993 for Papers 1–5, respectively. The lists of URs are presented in Appendix A.

### 3.2. Identification of Genes Associated with Vascular Dysfunction from GWAS Studies

Our next goal was to search for genes for which previous GWAS studies have identified variants and risk alleles that are associated with vascular dysfunction. To this aim, we searched the GWAS Catalog for the following traits: hypertension, atherosclerosis, and arterial stiffness. For hypertension, the initial search retrieved a total number of 575 associations. The search was subsequently refined in terms of excluding studies related to early-onset hypertension, pulmonary arterial hypertension, pseudotumor cerebri, treatment-resistant hypertension, preeclampsia, chemotherapy-induced hypertension, or hypertension risk in short sleep duration, leading to a final list of 461 associations, and 375 genes associated with hypertension across 33 GWAS studies with the following accession numbers: GCST000041, GCST000361, GCST000398, GCST000447, GCST000849, GCST000973, GCST001085, GCST001238, GCST001423, GCST002627, GCST003613, GCST004143, GCST004384, GCST004388, GCST006023, GCST006229, GCST007707, GCST008036, GCST008828, GCST009685, GCST010477, GCST010774, GCST011141, GCST011952, GCST011953, GCST011954, GCST012136, GCST90000060, GCST90000064, GCST90077646, GCST90086091, GCST90086092, GCST90086157. For atherosclerosis, the initial search retrieved a total number of 261 associations. The search was subsequently refined in terms of exclusion of a study on the interaction of traffic-related air pollution with peripheral arterial disease, leading to a final list of 102 associations, and 69 genes associated with atherosclerosis across 19 GWAS studies with the following accession numbers: GCST000720, GCST001231, GCST002504, GCST003154, GCST007425, GCST007435, GCST008474, GCST009134, GCST010549, GCST90013689, GCST90013731, GCST90018670, GCST90018890, GCST90043957, GCST90061371, GCST90061372, GCST90061374, GCST90061375, GCST90061376. For arterial stiffness, a total number of 62 associations and 58 genes were identified in six GWAS studies with the following accession numbers: GCST000370, GCST007846, GCST008403, GCST010654, GCST010655, GCST010656. Pulling together all of these genes, and after the removal of duplicates, we finally obtained a list of *n* = 493 genes that previous GWAS studies have associated with vascular dysfunction (Appendix A).

### 3.3. Integration of Transcriptomic Data with GWAS Identified Genes

Aiming to identify which of the DEGs from the human intervention studies selected for our analyses may potentially have the capacity to underlie the interindividual variability of the vascular effects in response to flavonoids intake, we conducted an integrative analysis of transcriptomic data and the genes identified from GWAS. To this end, for each of the selected studies, we compared the DEGs with the trait-specific genes identified from GWAS, i.e., genes whose variants are associated with hypertension, atherosclerosis, or arterial stiffness. For Paper 1, we identified 20, 4, and 2 genes associated with hypertension, atherosclerosis, or arterial stiffness, respectively; for Paper 2—13, 2, and 1 genes; for Paper 3—5, 1, and 4 genes; for Paper 4—33, 4 and 9 genes; and for Paper 5—18, 3, and 5 genes associated with hypertension, atherosclerosis, or arterial stiffness, respectively (Figure 3; Appendix A).

When pulling together all of these genes, we identified *n* = 106 DEGs that potentially have the capacity to underlie the interindividual variability of the vascular effects in response to flavonoids intake, as candidate genes for future nutrigenetic studies on flavonoids and vascular health (Appendix A), here referred to as *candidate DEGs*.

#### Functional Analysis of Candidate DEGs

To better understand the biological functions of identified candidate DEGs (*n* = 106) and prioritize some of them for subsequent analyses, we performed functional analyses by determining their place in canonical pathways using pathway enrichment analyses. These analyses pinpointed several pathways of relevance for vascular dysfunction. Some of these pathways are directly involved in vascular dysfunction such as the VEGFA-VEGFR2 signaling pathway, which contains six candidate DEGs, regulation of actin cytoskeleton (four candidate DEGs), adherens junction (three candidate DEGs), focal adhesion (three candidate DEGs), apelin signaling pathway (two candidate DEGs), composition of lipid particles (two candidate DEGs), fluid shear stress and atherosclerosis (two candidate DEGs), or platelet activation (two candidate DEGs), while others are involved in inflammation, cell signaling, or antioxidant protection, such as the chemokine signaling pathway, NF-kappa B signaling pathway, toll-like receptor signaling pathway, MAPK signaling pathway, PI3K-Akt signaling pathway, or the NRF2 pathway. All of these pathways and their associated candidate DEGs are presented in Table 2. In summary, with these pathway enrichment analyses, we identified *n* = 26 DEGs that are placed in KEGG pathways and *n* = 25 DEGs that are placed in WikiPathways. After the removal of duplicate genes, there were *n* = 33 candidate DEGs placed in KEGG or WikiPathways that are relevant to vascular dysfunction (Appendix A).

To identify genes with potentially greater influence on the interindividual variability of the vascular effects of flavonoids intake, and prioritize some of them for subsequent analyses, we searched for which of the candidate DEGs are among those that are common in the selected studies. To this end, we conducted a comparative analysis of the DEGs that at least two studies have in common (*n* = 720) and the candidate DEGs (*n* = 106) and obtained a list of *n* = 15 genes (*CAPZA1*, *FSTL4*, *GNA13*, *LSP1*, *MRPL23*, *MS4A4A*, *NCR3*, *NOL10*, *NUMB*, *SARM1*, *SH2B3*, *SYTL3*, *TCF20*, *ZMYM2*, *ZNF831*). These genes are presented in Figure 4. Of note, three of these genes (*GNA13*, *NCR3*, *SARM1*), are associated with pathways related to vascular dysfunction, which are presented in Appendix A. In addition, we also conducted a comparative analysis of the DEGs that three or more studies had in common (*n* = 67) and the candidate DEGs (*n* = 106) and, in the intersection of the Venn diagram, we obtained only one gene, that is *CAPZA1* (Figure 4).

This analysis also pinpointed two interesting candidate DEGs that are associated with two of the analyzed traits each, namely *CDKN2B-AS1* in Paper 4, which is associated with both hypertension and atherosclerosis, and *HLA-DRB1* in Paper 5, which is associated with both atherosclerosis and arterial stiffness (Appendix A).

### 3.4. Identification of Candidate Genes for Nutrigenetic Studies among the URs of DEGs

After having identified the URs of DEGs for each of the selected studies (Appendix A), we aimed to identify for which of these regulators the previous GWAS studies had identified variants and risk alleles that are associated with vascular dysfunction. To this aim, we conducted comparative analyses between all GWAS genes and URs for each study separately, and identified *n* = 3 candidate URs for Publication 1 (*APOE*, *EBF1*, *ZBTB10*), *n* = 6 for Publication 2 (*H19*, *KPNA2*, *LDLR*, *PBRM1*, *TNF*, *ZNF746*), *n* = 7 for Publication 3 (*ERAP1*, *FMN2*, *FOXO1*, *HIC1*, *RPTOR*, *TCF7L2*, *ZNF746*), *n* = 25 for Publication 4 (*APOE*, *ARNTL*, *CERS5*, *CXCL8*, *CYP11B2*, *EBF1*, *EDNRA*, *FOXC1*, *FOXO1*, *GSE1*, *IL6*, *IRF5*, *MKNK1*, *MMP3*, *MYO9B*, *NFATC2*, *NPAS3*, *PLCE1*, *PPARG*, *PTPN11*, *SMARCA4*, *TCF20*, *TCF7L2*, *TNF*, *ZBTB10*), and *n* = 22 for Publication 5 (*APOE*, *EBF1*, *FOXC1*, *IL6*, *IRF5*, *KPNA2*, *LDLR*, *LSP1*, *MYO6*, *NCOR2*, *NCR3*, *NFATC2*, *NR1H3*, *PNPT1*, *PPARG*, *PRDM16*, *PTPN11*, *RPTOR*, *SMARCA4*, *TNF*, *USP8*, *ZBTB10*). After pulling together all of these genes and removing duplicates, we obtained *n* = 42 URs of flavonoid-modulated genes as potential candidate genes for future nutrigenetic studies on vascular effects of these bioactives (Appendix A), here referred to as *candidate URs*. Notably, the expression of some of these candidate URs is modulated with respective flavonoid interventions, such as: *KPNA2* for Publication 2, *ARNTL*, *MKNK1*, *MYO9B*, *TCF20* for Publication 4, and *IRF5*, *NCR3* for Publication 5.

### 3.5. Candidate Genes (DEGs and URs) in Published Studies on Genetic Polymorphisms, Cardiovascular Diseases, and Nutrition

To further prioritize candidate genes for subsequent analyses, we conducted a literature search for published studies that showed associations of these genes’ polymorphisms with cardiovascular diseases and atherosclerosis. For this purpose, in mid-August 2022, we searched PubMed applying the following wording: ((XXX[Title/Abstract]) AND ((SNP[Title/Abstract] OR polymorphism[Title/Abstract]))) AND (cardiovascular[Title/Abstract] OR atherosclerosis[Title/Abstract]), where “XXX” was replaced by the gene of interest. This literature search showed that among candidate DEGs, the number of publications was the largest for *MTHFR* (*n* = 394), *LPL* (*n* = 92), *LDLR* (*n* = 74), *ALDH2* (*n* = 47), and *TCF7L2* (*n* = 34), while among candidate URs, the number of publications was the largest for *APOE* (*n* = 383), *IL6* (*n* = 233), *TNF* (*n* = 140), *LDLR* (*n* = 74), *MMP3* (*n* = 46), *CYP11B2* (*n* = 45), *TCF7L2* (*n* = 34), and *PPRG* (*n* = 24).

Furthermore, we also conducted a PubMed search to identify previously published studies that showed associations of candidate genes’ polymorphisms with diet and nutrition. For this purpose, we applied the following wording: ((SNP[Title/Abstract]) AND ((nutrition[Title/Abstract] OR nutrient[Title/Abstract] OR diet[Title/Abstract]))) AND (XXX[Title/Abstract]), where “XXX” was replaced by the gene of interest. This literature search showed that among candidate DEGs, the number of publications was the largest for *MTHFR* (*n* = 22), *LPL* (*n* = 14), *LDLR* (*n* = 5), and *TCF7L2* (*n* = 5), while among candidate URs, the number of publications was the largest for *APOE* (*n* = 26), *IL6* (*n* = 13), *TNF* (*n* = 10), *PPARG* (*n* = 6), *LDLR* (*n* = 5), and *TCF7L2* (*n* = 5).

### 3.6. Selection of Top-Priority Candidate Genes and Their Polymorphisms Potentially Associated with Flavonoids and Vascular Health

Our next step was directed towards final prioritization of a subset of candidate genes. To this end, using the results from the above analyses, we conducted the following additional analyses: (a) intersection between candidate DEGs in two or more papers AND candidate DEGs in canonical pathways; (b) intersection between candidate DEGs in two or more papers AND candidate URs; (c) intersection between candidate DEGs AND top genes in published studies on genetic polymorphisms, cardiovascular diseases, and nutrition; (d) intersection between candidate URs AND top genes in published studies on genetic polymorphisms, cardiovascular diseases, and nutrition; (e) intersection between candidate DEGs in canonical pathways AND candidate URs; (f) intersection between candidate DEGs AND DEGs in three or more papers (i.e., candidate DEGs in three or more papers). The rationale behind conducting these analyses is the following: If a candidate gene takes some of the central positions within these integrative analyses, it is much more likely that it will significantly influence the vascular effects of flavonoids. These analyses allowed us to select *n* = 20 top-priority candidate genes: *ALDH2*, *APOE*, *CAPZA1*, *CYP11B2*, *GNA13*, *IL6*, *IRF5*, *LDLR*, *LPL*, *LSP1*, *MKNK1*, *MMP3*, *MTHFR*, *MYO6*, *NCR3*, *PPARG*, *SARM1*, *TCF20*, *TCF7L2*, and *TNF*.

For each of these top-priority candidate genes, we interrogated the Variation Viewer database to identify their variant types, molecular consequences, most severe clinical significances, and top 10 genetic variants with highest frequencies, which are presented in Table 3. Also, for each of these genes, we interrogated the PharmGKB database with the aim of identifying variants that have already been shown to have specific pharmacologic relevance, and the number of associations is also presented in Table 3.

In addition, GWAS-reported variants of selected top-priority candidate genes associated with vascular dysfunction were identified back in the GWAS Catalog, while their frequencies in the global population were identified in the dbSNP database (Table 4). In the same database, we also searched for the previously reported clinical significance of each of these variants and found that rs671 (*ALDH2*) is a risk factor, pathogenic, drug response, and protective; rs6418 (*CYP11B2* and also *GML*) is benign; rs1799998 (*CYP11B2* and also *LY6E-DT*) has association and is benign; rs6511720 (*LDLR*) is benign; rs17367504 (*MTHFR*) is benign; rs7903146 (*TCF7L2*) is a likely risk allele and risk factor. For other genetic variants included in Table 4, no clinical significance was reported in the dbSNP database. In addition, the role of each candidate gene, whether it is a DEG, UR, or both, is reported in Table 4.

## 4. Discussion

Gene–diet interaction has long been considered one of the key determinants of interindividual variability in the effect of a number of dietary factors [44]. Among the classic examples are studies on the interactions between genetic polymorphism, intake of bioactives related to coffee consumption and the risk of acute myocardial infarction. These studies identified caffeine as a key factor for increased risk only for individuals with slow caffeine metabolism [45] that is associated with the −163A > C (rs762551) single nucleotide polymorphism of the Cytochrome P450 Family 1 Subfamily A Member 2 (*CYP1A2*) gene. This SNP has been shown to alter the inducibility and activity of the CYP1A2 enzyme, which accounts for approximately 95% of caffeine metabolism in the body. Individuals with the AC or CC genotype are categorized as slow metabolizers, while individuals with the AA genotype are categorized as fast metabolizers [46]. Furthermore, a recent study has shown an increased risk of hypertension and renal dysfunction with heavy coffee intake, but only among individuals with the AC and CC genotypes of *CYP1A2* at rs762551 [47]. Consequently, an influence of genetic polymorphisms on the vasculoprotective properties of dietary flavonoids can also be expected, as one of the determinants of interindividual variability in the effect. A recent study identified for the first time genetic polymorphisms that determine the effect of orange juice consumption on circulating lipids and blood pressure. In the study group of 46 participants, medium or high excretors of flavanone metabolites, it was observed that for the *APOA1*_rs964184 polymorphism, the CC genotype is associated with a decrease in circulating triglycerides and blood pressure, both systolic and diastolic. Additionally, for the *LPL*_rs12678919 polymorphism, the AA genotype was associated with a change in blood lipids [20].

Given that vascular health is governed not only by genes directly associated with vascular tone, vascular permeability, and circulating lipoproteins, but also by genes involved in general metabolic dysregulation, it is realistic to expect that a greater number of genetic polymorphisms determine the interindividual variabilities of the vascular health effects of flavonoids, which highlights the need for their identification and further nutrigenetic studies. To address this issue, GWAS represent a valuable source of information, the approach of which involves genome-wide analysis of genotypes of a large number of individuals to identify variants associated with a specific disease or health-related trait compared to healthy individuals, i.e., identification of genotype–phenotype associations. So far, GWAS have identified hundreds of genetic variants that are associated with different diseases or health-related traits in humans [48]. More importantly, the data from numerous GWAS analyses are aggregated, structured, and standardized into a publicly accessible database [49], allowing them to be utilized in future research. A major limitation of GWAS is that they only provide a statistical association between a specific genetic variant and a given disease or trait. In other words, GWAS provide genes associated with specific diseases or traits and do not necessarily pinpoint causal variants and genes [48]. Understanding potential functional consequences of identified variants represents a considerable challenge, for which various approaches have been proposed [21,22,23,24,50,51]. One approach proposed in a recent study consisted of integrating GWAS and mRNA microarray data to computationally identify key disease pathways, upstream regulators, and downstream therapeutic targets in primary biliary cholangitis [52]. Specifically, GWAS analysis conducted on 1920 patients and 1770 healthy controls identified 261 genes associated with primary biliary cholangitis, in parallel to mRNA microarray analysis that was conducted in liver needle biopsy specimens from 36 patients and 5 controls and identified 1574 DEGs. Subsequent functional analyses, which included signaling networks analyses and analyses of upstream regulators, enabled the prediction of central regulators in disease susceptibility and identified potential downstream therapeutic targets [52].

To address our aim, which is to identify candidate genes which polymorphisms potentially determine the interindividual variabilities in the effects of flavonoids on vascular health, we employed an integrative analysis of GWAS (i.e., genetic) and mRNA microarray (i.e., genomic) data using (a) available global transcriptomic data of published human intervention studies on flavonoids and vascular health demonstrating a positive effect on vascular function and (b) genes associated with vascular dysfunction-related traits (hypertension, atherosclerosis, and arterial stiffness) identified from GWAS. In addition, for each set of genomic data, we identified the URs, molecules capable of regulating the expression of DEGs. Some of these URs are proteins, which are inherently prone to genetic variability, thus potentially serving as a source for significant interindividual variabilities in flavonoids and vascular health. Even though flavonoids are a large and diverse class of (poly)phenols, previous transcriptomic studies have shown that not only (poly)phenols from specific classes, but (poly)phenols in general can exhibit common molecular mechanisms of action [53], most likely because some of them are metabolized by gut microbiota to similar or identical metabolites that mediate their molecular mechanisms of action [54], which was the rationale behind our decision to include human intervention studies conducted with different dietary flavonoids. By employing integrative analysis of transcriptomic and GWAS datasets, we added an important step in the identification of candidate genes for future nutrigenetic studies. This integrative analysis identified 106 *candidate DEGs* and 42 *candidate URs*. Subsequent functional analyses and a literature search of these candidate DEGs and URs identified 20 top-priority candidates: *ALDH2*, *APOE*, *CAPZA1*, *CYP11B2*, *GNA13*, *IL6*, *IRF5*, *LDLR*, *LPL*, *LSP1*, *MKNK1*, *MMP3*, *MTHFR*, *MYO6*, *NCR3*, *PPARG*, *SARM1*, *TCF20*, *TCF7L2*, and *TNF*. It should be added that our study only focused on genes directly associated with the vascular effects of flavonoids and did not consider genes involved in their absorption and metabolism. Therefore, the results of our study should be verified by carefully designed human nutrigenetic studies that will only include individuals with high levels of circulating metabolites of the tested flavonoids. Such an approach would eliminate the influence of interindividual variabilities in the absorption and metabolism of flavonoids, phenomena that are well-identified but still poorly understood.

Among the top candidate genes and their known SNPs, there is evidence about their functionality related to vascular health. For example, the rs7903146 variant of the Transcription Factor 7 Like 2 (*TCF7L2*) gene was identified as associated with type 2 diabetes [55], and the T allele of this variant strongly predicts future type 2 diabetes. This allele is associated with enhanced expression of *TCF7L2* in human islets as well as with impaired insulin secretion [56]. Furthermore associations between rs7903146 and (a) elevated serum triglycerides in patients with familial combined hyperlipidemia [57], (b) impaired postprandial lipid metabolism in healthy young males and elderly persons [58], (c) inflammation, metabolic dysregulation, and atherosclerotic cardiovascular diseases [59] were observed. TCF7L2 is a transcription factor and the ultimate effector of the Wnt signaling pathway, which plays an important protective role in the development of atherosclerotic cardiovascular diseases [59]. Regarding the results from global gene expression studies on flavonoids and vascular function, *TCF7L2* has been identified as a differentially expressed gene in one study and as an upstream regulator in two studies. These observations suggest that the *TCF7L2* gene is one of the potential key mediators of the interindividual differences to flavonoid intake.

Another polymorphism with proven functionality in vascular dysfunction is rs6511720 of the Low-Density Lipoprotein Receptor (*LDLR*) gene, for which a recent study has shown a significant association with susceptibility to coronary artery disease, as well as with regression of carotid intima-media thickness and changes in plasma lipids during rosuvastatin therapy [60]. A single-nucleotide polymorphism, rs1799998, in the aldosterone synthase gene, Cytochrome P450 Family 11 Subfamily B Member 2 (*CYP11B2*), has also been reported to associate with cardiovascular diseases, such as atrial fibrillation [61] or intracranial large artery stenosis [62]. In addition, it has been shown that this polymorphism is associated with a predisposition to the development of late in-stent restenosis in heterozygous patients with stable coronary artery disease [63]. Other polymorphisms that have not only been statistically associated with vascular dysfunction but have also been functionally related to it include Aldehyde Dehydrogenase 2 Family Member (*ALDH2*) rs671 and Methylenetetrahydrofolate Reductase (*MTHFR*) rs17367504. These genes are crucial in alcohol metabolism and folate/homocysteine metabolism, respectively. The rs671 polymorphism in *ALDH2* was pinpointed as a risk factor for the occurrence of death from cardio-cerebrovascular complications in patients with type 2 diabetes [64] and has recently been characterized as a novel regulator of cholesterol biosynthesis [65]. For the *MTHFR* rs17367504, it was not only associated with hypertension in a previous GWAS but was also included in the calculation of genetic risk score (GRS) in a study aiming to evaluate whether the association between GRS and blood pressure was modified by usual coffee consumption. This study revealed that individuals with greater GRS present high blood pressure associated with higher coffee consumption, highlighting the particular importance of reducing coffee intake in individuals who are genetically predisposed to this cardiovascular disease risk factor [66]. Moreover, polymorphisms in *LPL* and *APOE* genes were suggested to modulate the effects of orange juice rich in flavanone on vascular function [20]. Taken together, these examples strongly suggest that the candidate genes identified using our integrative analyses of genomic and GWAS data are good candidates, demonstrating the power of such an approach for the identification of novel, still unexplored candidate genes involved in interindividual variability in response to flavonoid intake and vascular health.

There are several limitations to this study. The major limitation is the small number of genomic datasets used, five, corresponding to the only studies available that aimed to assess global genomic change induced by flavonoids in human volunteers associated with positive vascular health effects. Also, the number of studies is limited as we only used genomic data that were obtained using a microarray approach. In addition, recent studies have revealed that (poly)phenols exert their health effects by modulating the expression of not only protein coding genes but also the expression of protein non-coding genes, such as microRNAs or long non-coding RNAs [53,67], and by exerting changes in the DNA methylation profile [68]. Also, GWAS have identified vascular dysfunction-associated variants in the non-coding elements of the genome [69]. Therefore, integration of genomic data with GWAS data in future analyses should include information about changes in the expression of all types of RNA as well as the DNA methylation profile.

## 5. Conclusions

In conclusion, we performed an integrated bioinformatic analysis with large-scale GWAS and transcriptomic data to generate a refined list of candidate causal genes for interindividual variability in response to flavonoid intake. These results should serve as an important resource, facilitating the focusing of nutrigenetic research in the field of plant food bioactives to identify gene variants associated with a better health response to these bioactives, and therefore build a foundation for precision nutrition research in the field of (poly)phenols.

## Figures and Tables

**Figure 1 nutrients-16-01362-f001:**
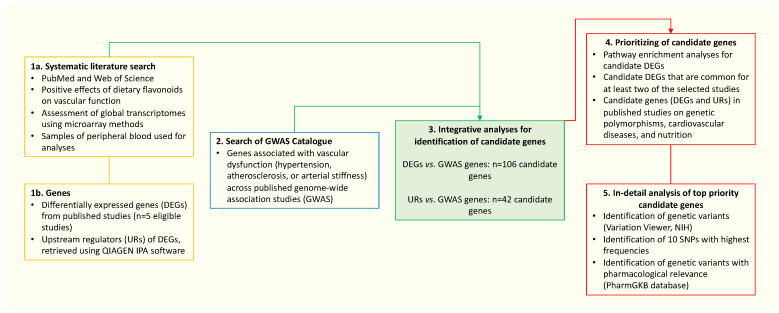
Flowchart of the study.

**Figure 2 nutrients-16-01362-f002:**
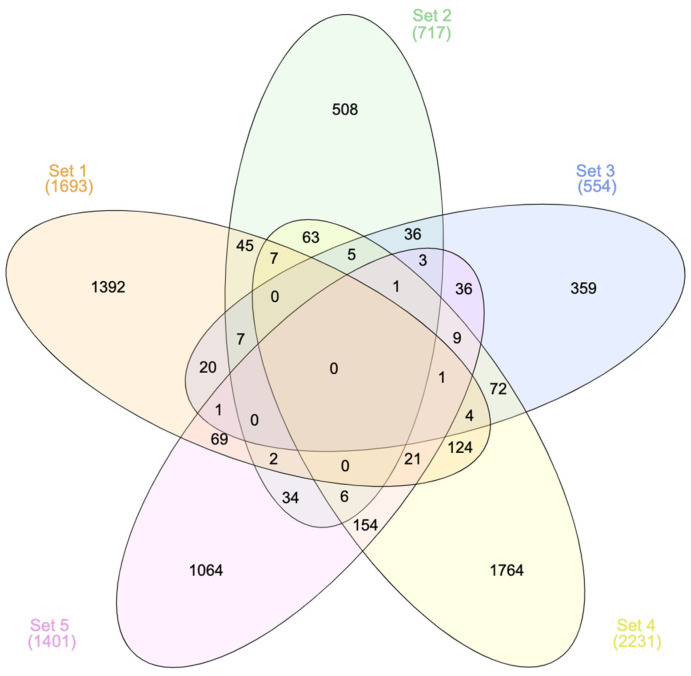
Venn diagram representing the number of common genes across the selected studies. Sets 1–5 refer to the sets of DEGs extracted from Papers 1–5, respectively. General information about the papers (Paper 1 [26]; Paper 2 [27]; Paper 3 [28]; Paper 4 [30]; Paper 5 [29]) is presented in Table 1. The online tool InteractiVenn was used for conducting the analysis and visualizing the results.

**Figure 3 nutrients-16-01362-f003:**
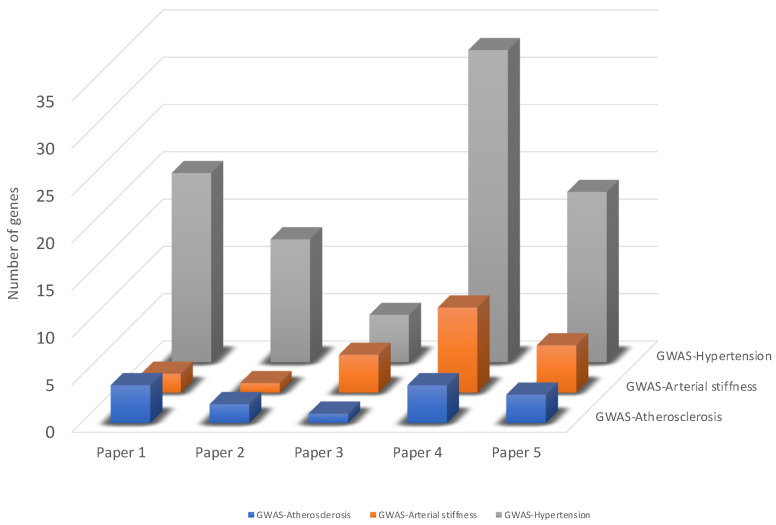
Comparative analysis of DEGs and trait-specific genes identified from GWAS. For each of the selected papers (Papers 1–5), DEGs were compared with trait-specific genes identified from GWAS, i.e., genes whose variants are associated with hypertension, atherosclerosis, or arterial stiffness (Appendix A). Papers 1–5 refer to the papers included in this integrative analysis. General information about the papers (Paper 1 [26]; Paper 2 [27]; Paper 3 [28]; Paper 4 [30]; Paper 5 [29]) is presented in Table 1.

**Figure 4 nutrients-16-01362-f004:**
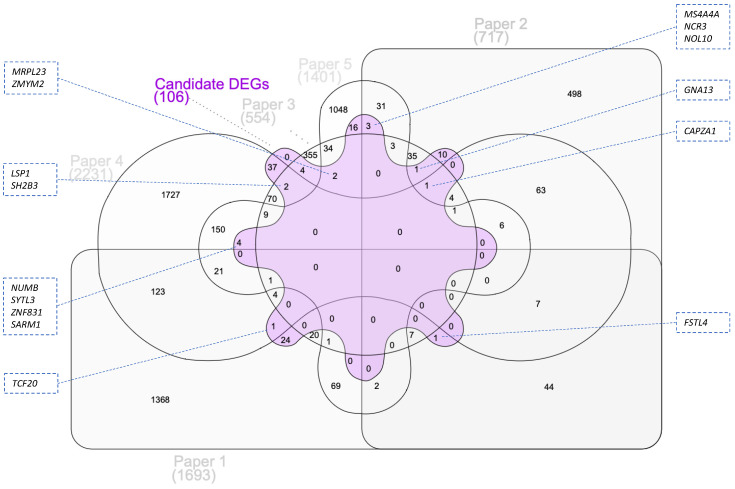
Identification of candidate DEGs which are among those that are common in the selected studies. A comparative analysis of DEGs that at least two studies have in common (*n* = 720) and candidate DEGs (*n* = 106) retrieved *n* = 15 genes with potentially greater influence on interindividual variability of the vascular effects of flavonoids intake: *CAPZA1*, *FSTL4*, *GNA13*, *LSP1*, *MRPL23*, *MS4A4A*, *NCR3*, *NOL10*, *NUMB*, *SARM1*, *SH2B3*, *SYTL3*, *TCF20*, *ZMYM2*, *ZNF831*. Papers 1–5 refer to the papers included in this integrative analysis. General information about these papers (Paper 1 [26]; Paper 2 [27]; Paper 3 [28]; Paper 4 [30]; Paper 5 [29]) is presented in Table 1. The online tool InteractiVenn was used for conducting the analysis and visualizing the results.

**Table 1 nutrients-16-01362-t001:** Human intervention studies (Papers 1–5) included in this integrative analysis, and associated papers reporting vascular function-related outcomes.

Paper	Title and Reference	Study Population	Bioactives	Outcomes	Associated Paper for Outcomes
1.	Hesperidin displays relevant role in the nutrigenomic effect of orange juice on blood leukocytes in human volunteers: a randomized controlled cross-over study [26]	Healthy, middle-aged, moderately overweight men	Hesperidin	Decreased diastolic blood pressure	Hesperidin contributes to the vascular protective effects of orange juice: a randomized crossover study in healthy volunteers [40]
2.	Dietary flavanols modulate the transcription of genes associated with cardiovascular pathology without changes in their DNA methylation state [27]	Non-obese, healthy male smokers, smoking 10 and more cigarettes per day for at least 5 years	Monomeric and oligomeric flavanols from grape seeds	Improved vascular health index	Pleiotropic benefit of monomeric and oligomeric flavanols on vascular health: a randomized controlled clinical pilot study [41]
3.	Circulating anthocyanin metabolites mediate vascular benefits of blueberries: insights from randomized controlled trials, metabolomics, and nutrigenomics [28]	Healthy male volunteers	Wild blueberry anthocyanins	Increased flow-mediated vasodilatation Decreased 24 h systolic blood pressure	/
4.	Flavanol consumption in healthy men preserves integrity of immunological–endothelial barrier cell functions: nutri(epi)genomic analysis [30]	Healthy middle-aged men	Cocoa flavanols	Increased flow-mediated vasodilatation Decreased systolic and diastolic blood pressure Decreased pulse wave velocity	Cocoa flavanol intake improves endothelial function and Framingham Risk Score in healthy men and women: a randomised, controlled, double-masked trial: the Flaviola Health Study [42]
5.	Grapefruit juice flavanones modulate the expression of genes regulating inflammation, cell interactions and vascular function in peripheral blood mononuclear cells of postmenopausal women [29]	Healthy, non-smoking women, 3 to 10 years after menopause	Grapefruit juice flavanones	Decreased carotid–femoral pulse wave velocity	Flavanones protect from arterial stiffness in postmenopausal women consuming grapefruit juice for 6 mo: a randomized, controlled, crossover trial [43]

Papers 1–5 report human intervention studies with flavonoids demonstrating beneficial effects on vascular function together with modulations in global gene expression in peripheral blood cells.

**Table 2 nutrients-16-01362-t002:** Functional analysis of candidate DEGs: candidate DEGs that are placed in canonical pathways relevant for vascular dysfunction.

Canonical Pathways	Number of Hits	Genes
** *Pathways directly involved in vascular dysfunction* **		
VEGFA-VEGFR2 Signaling Pathway *	6	*CSK*, *MKNK1*, *MYO6*, *PTPRJ*, *SMARCA2*, *TNXB*
Regulation of actin cytoskeleton **	4	*BRK1*, *CSK*, *GNA13*, *SOS2*
Adherens junction	3	*PTPRJ*, *TCF7L2*, *YES1*
Angiopoietin Like Protein 8 Regulatory Pathway *	3	*LPL*, *PRKAG2*, *SOS2*
ECM-receptor interaction	3	*NPNT*, *TNXB*, *VTN*
Focal adhesion	3	*SOS2*, *TNXB*, *VTN*
Apelin signaling pathway	2	*GNA13*, *PRKAG2*
Cholesterol metabolism	2	*LDLR*, *LPL*
Composition of Lipid Particles *	2	*LDLR*, *LPL*
Fluid shear stress and atherosclerosis	2	*BMPR1B*, *GSTA4*
Glycerolipid metabolism	2	*ALDH2*, *LPL*
Metabolic pathway of LDL, HDL and TG, including diseases *	2	*LDLR*, *LPL*
Platelet activation	2	*GNA13*, *LYN*
Statin Pathway *	2	*LDLR*, *LPL*
** *Pathways involved in inflammation* **		
Chemokine signaling pathway **	3	*CSK*, *LYN*, *SOS2*
Cytokine-cytokine receptor interaction	3	*BMPR1B*, *GDF10*, *LTB*
NF-kappa B signaling pathway **	3	*LTB*, *LYN*, *TAB2*
Regulation of toll-like receptor signaling pathway *	3	*IRF5*, *SARM1*, *TAB2*
B cell receptor signaling pathway	2	*LYN*, *SOS2*
Interleukin-11 Signaling Pathway *	2	*FES*, *YES1*
Natural killer cell mediated cytotoxicity	2	*NCR3*, *SOS2*
Structural Pathway of Interleukin 1 (IL-1) *	2	*MKNK1*, *TAB2*
TNF signaling pathway	2	*DAB2IP*, *TAB2*
Toll-like receptor signaling pathway **	2	*IRF5*, *TAB2*
** *Cell signaling pathways* **		
MAPK signaling pathway **	4	*MKNK1*, *SOS2*, *STK3*, *TAB2*
PI3K-Akt signaling pathway **	4	*MCL1*, *SOS2*, *TNXB*, *VTN*
EGF/EGFR Signaling Pathway *	3	*CSK*, *SOS2*, *TWIST1*
Insulin signaling pathway	3	*MKNK1*, *PRKAG2*, *SOS2*
Sterol Regulatory Element-Binding Proteins (SREBP) signalling *	3	*LDLR*, *LPL*, *PRKAG2*
cGMP-PKG signaling pathway	2	*ATP2B1*, *GNA13*
FoxO signaling pathway	2	*PRKAG2*, *SOS2*
Jak-STAT signaling pathway	2	*MCL1*, *SOS2*
Phospholipase D signaling pathway	2	*GNA13*, *SOS2*
Wnt Signaling Pathway and Pluripotency *	2	*LDLR*, *TCF7L2*
** *Antioxidant protection* **		
NRF2 pathway *	2	*GSTA4*, *SLC39A8*

Legend: No asterisk—KEGG pathways; * WikiPathways; ** both KEGG and WikiPathways.

**Table 3 nutrients-16-01362-t003:** Variant types, molecular consequences, most severe clinical significances, and top 10 genetic variants with highest frequencies, as well as the number of associations with pharmacologic relevance in selected *n* = 20 top-priority candidate genes.

Gene	Variant Type (Number)	Molecular Consequences (Number)	Most Severe Clinical Significance (Number)	Number of Associations with Pharmacological Significance	10 Variants with Highest Frequencies				
					Variant ID	Molecular Consequences	Alleles	Alleles with Highest Frequencies	Frequency
*ALDH2*	single nucleotide variant (172) deletion (3) insertion (2) indel (22)	missense variant (1) intron variant (131) 3 prime UTR variant (27) 500 B downstream variant (1) 2 KB upstream variant (14)	pathogenic (1)	9	rs7296651 rs6489793 rs2106697 rs10774638 rs886205 rs4767939 rs10774637 rs9971942 rs10774639 rs11066028	intron variant 3 prime UTR variant not specified intron variant 2 KB upstream variant intron variant intron variant not specified not specified intron variant	C,A,G T,G T,A,C,G T,A,C A,C,G,T A,G C,G,T C,T G,A,C,T A,C,G,T	C T T T A G T T A A	0.497804 0.497604 0.496406 0.492612 0.491214 0.419728 0.419129 0.410743 0.410743 0.372804
*APOE*	single nucleotide variant (8)	missense variant (3) synonymous variant (1) intron variant (4) 5 prime UTR variant (1) 500 B downstream variant (1) 2 KB upstream variant (1)	pathogenic (2) drug-response (2)	33	rs405509 rs440446 rs769450 rs429358 rs7412 rs769449 rs1081105 rs877973	2 KB upstream variant missense variant, intron variant, synonymous variant intron variant missense variant missense variant intron variant 500 B downstream variant 5 prime UTR variant, intron variant	T,G C,G,T G,A T,C C,T G,A A,C,G C,A,T	T C A C T A C A	0.471845 0.373802 0.327276 0.150559 0.0750799 0.0648962 0.0301518 0.0159744
*CAPZA1*	single nucleotide variant (136) deletion (1) insertion (1) indel (26)	missense variant (1) synonymous variant (2) intron variant (149) nc transcript variant (1) 5 prime UTR variant (3) 3 prime UTR variant (1) 500 B downstream variant (2) 2 KB upstream variant (22)	/	/	rs3013440 rs7524494 rs7415820 rs3103450 rs2932536 rs3013439 rs12046329 rs12046466 rs12046208 rs9429486	intron variant intron variant intron variant intron variant intron variant intron variant intron variant intron variant intron variant intron variant	G,A A,G,T G,A G,A,T G,A,T T,A,G T,A,C,G T,A,C C,T T,A,C,G	G A G G G T T T T T	0.478435 0.478435 0.478035 0.478035 0.478035 0.477835 0.477835 0.477835 0.436502 0.435703
*CYP11B2*	single nucleotide variant (86) insertion (2) indel (3)	missense variant (9) synonymous variant (6) intron variant (66) 3 prime UTR variant (7) 500 B downstream variant (2) 2 KB upstream variant (5)	pathogenic (1) association (1) benign (17)	3	rs10110732 rs28615142 rs28366703 rs13263682 rs6421 rs79201878 rs80062072 rs74838461 rs28394055 rs6429	intron variant intron variant intron variant intron variant intron variant intron variant intron variant intron variant intron variant intron variant	C,T T,C A,G A,C T,C A,T A,G T,C C,T G,A,C,T	C C G C C T G C T C	0.495208 0.483227 0.482628 0.479433 0.460064 0.45607 0.45607 0.45607 0.445887 0.44389
*GNA13*	single nucleotide variant (164) deletion (2) insertion (1) indel (18)	missense variant (2) synonymous variant (1) intron variant (133) 5 prime UTR variant (2) 3 prime UTR variant (11) 500 B downstream variant (1) 2 KB upstream variant (6)	/	/	rs9911189 rs2011307 rs6504271 rs12944877 rs12939956 rs7501452 rs3960369 rs8082708 rs12945514 rs4791243	intron variant intron variant intron variant intron variant not specified not specified not specified intron variant intron variant intron variant	A,G C,T C,T G,A,C,T C,T T,C C,A,G,T T,C G,C,T T,A,C,G	A C C G C T C T G T	0.440096 0.244209 0.244209 0.238618 0.236621 0.235024 0.234026 0.233826 0.228235 0.228035
*IL6*	single nucleotide variant (38) insertion (1) indel (3)	missense variant (4) synonymous variant (2) intron variant (27) nc transcript variant (5) 3 prime UTR variant (6) 500 B downstream variant (10) 2 KB upstream variant (16)	risk-factor (1) benign (2)	11	rs7802308 rs34328912 rs1800796 rs1524107 rs2066992 rs2069845 rs1554606 rs2069840 rs1474347 rs367801961	intron variant 500 B downstream variant nc transcript variant, intron variant intron variant, 2 KB upstream variant intron variant, 2 KB upstream variant 3 prime UTR variant, intron variant intron variant, 2 KB upstream variant intron variant, 2 KB upstream variant intron variant, 2 KB upstream variant 500 B downstream variant	T,A A,C,T G,A,C C,G,T G,A,C,T G,A,C,T T,A,G C,G C,A,G G,A,T	A C C T T G T G C A	0.454473 0.336062 0.313898 0.308307 0.308107 0.252596 0.249401 0.185503 0.168331 0.167133
*IRF5*	single nucleotide variant (59) indel (6)	intron variant (44) splice donor variant (1) 5 prime UTR variant (8) 3 prime UTR variant (6) 500 B downstream variant (3) 2 KB upstream variant (4)	risk-factor (2) benign (1)	/	rs3757385 rs3807135 rs752637 rs3757388 rs10954213 rs13242262 rs7808907 rs11770589 rs1874327 rs10954214	5 prime UTR variant intron variant intron variant 2 KB upstream variant 3 prime UTR variant not specified intron variant 3 prime UTR variant intron variant 3 prime UTR variant	T,G T,C T,A,C G,A G,A A,G,T T,C G,A,C,T A,C,T C,T	T T T A G A C A A C	0.497604 0.49361 0.491014 0.471845 0.464058 0.463858 0.460264 0.453474 0.400559 0.399561
*LDLR*	single nucleotide variant (297) deletion (1) indel (41)	missense variant (4) nonsense (stop gained) (1) synonymous variant (8) intron variant (232) nc transcript variant (41) 3 prime UTR variant (28) 500 B downstream variant (11) 2 KB upstream variant (19)	pathogenic (1) conflicting-interpretations-of-pathogenicity (1) uncertain-significance (2) likely-benign (8) benign-likely-benign (7) benign (30)	5	rs17248931 rs73017023 rs73017025 rs4804145 rs8102912 rs2738459 rs10422256 rs3180023 rs2738456 rs2738458	intron variant intron variant intron variant intron variant intron variant intron variant intron variant nc transcript variant, 3 prime UTR variant intron variant intron variant	G,A,C A,T A,G G,A G,A,C A,C,G,T G,A,C,T C,A,G,T T,A,C T,C,G	A T G A A A A C C C	0.453674 0.453474 0.453275 0.451278 0.446486 0.425519 0.421526 0.41274 0.40655 0.40635
*LPL*	single nucleotide variant (196) deletion (2) indel (18)	missense variant (1) nonsense (stop gained) (1) synonymous variant (3) intron variant (161) 3 prime UTR variant (16) 500 B downstream variant (6) 2 KB upstream variant (9)	other (1) association (2) likely-benign (1) benign (21)	3	rs253 rs1534649 rs10104051 rs2197089 rs258 rs285 rs314 rs301 rs326 rs56321069	intron variant intron variant intron variant not specified intron variant intron variant intron variant intron variant intron variant intron variant	C,G,T G,A,T C,T G,A G,A,C,T C,G,T G,A T,C A,G T,A,G	C T T G G C A C G A	0.487819 0.480032 0.476038 0.457867 0.440695 0.439297 0.390176 0.381989 0.349441 0.346246
*LSP1*	single nucleotide variant (269) deletion (2) insertion (1) indel (18)	missense variant (4) synonymous variant (2) intron variant (238) nc transcript variant (1) 5 prime UTR variant (10) 3 prime UTR variant (2) 500 B downstream variant (8) 2 KB upstream variant (42)	/	/	rs7396311 rs810021 rs28971510 rs1092608 rs542605 rs10734623 rs517101 rs3781961 rs3817197 rs7122680	not specified not specified not specified not specified intron variant intron variant intron variant intron variant intron variant intron variant	A,G G,A,C,T G,A,C,T T,A,G A,C,G,T C,T A,G,T C,G,T G,A,C G,A,C,T	G G G T A T A C G C	0.499002 0.482628 0.48143 0.480831 0.479633 0.475639 0.470647 0.449081 0.447484 0.447284
*MKNK1*	single nucleotide variant (200) deletion (2) indel (28)	missense variant (2) synonymous variant (4) intron variant (206) nc transcript variant (14) 5 prime UTR variant (35) 3 prime UTR variant (6) 500 B downstream variant (9) 2 KB upstream variant (17)	/	/	rs11211303 rs3766243 rs7543083 rs1258049 rs2181414 rs3766240 rs11211319 rs11211320 rs12022855 rs12136479	intron variant intron variant intron variant intron variant intron variant intron variant, 500 B downstream variant intron variant intron variant intron variant intron variant	T,A,C A,G,T G,A G,A,T T,A,C G,A,C,T T,A,C T,C,G C,T A,C,G,T	C T A A T G T T C A	0.496006 0.49381 0.49361 0.479633 0.472244 0.446086 0.413339 0.413339 0.390375 0.380791
*MMP3*	single nucleotide variant (37) deletion (1) indel (2)	missense variant (2) nonsense (stop gained) (1) synonymous variant (4) intron variant (32) nc transcript variant (3) 3 prime UTR variant (2) 500 B downstream variant (1) 2 KB upstream variant (3)	benign (6)	4	rs650108 rs538161727 rs639752 rs602128 rs575027 rs520540 rs678815 rs591058 rs679620 rs617819	intron variant intron variant nc transcript variant, intron variant missense variant, synonymous variant intron variant, 500 B downstream variant synonymous variant intron variant intron variant missense variant, nonsense (stop gained) 2 KB upstream variant	G,A,T C,A,T C,A,T A,C,G A,C,G,T A,G,T G,A,C,T T,A,C T,A,C,G C,A,G	A T C A A A G T T C	0.442692 0.390974 0.385383 0.38139 0.378994 0.377596 0.35643 0.355232 0.347843 0.347244
*MTHFR*	single nucleotide variant (133) insertion (1) indel (17)	missense variant (13) nonsense (stop gained) (1) synonymous variant (7) intron variant (134) nc transcript variant (10) 5 prime UTR variant (10) 3 prime UTR variant (23) 500 B downstream variant (5) 2 KB upstream variant (24)	pathogenic (1) likely-pathogenic (1) other (1) uncertain-significance (1) benign (5)	139	rs10864543 rs4846052 rs6541005 rs3737966 rs1994798 rs7526128 rs6541003 rs4846049 rs11586659 rs2151655	synonymous variant intron variant intron variant 3 prime UTR variant, intron variant intron variant intron variant intron variant 3 prime UTR variant, 500 B downstream variant intron variant intron variant, 500 B downstream variant	C,G,T T,A,C A,T C,A,G,T G,A C,A,G,T G,A,C T,A,G T,A,C,G G,A,C,T	T T A C G C G T T G	0.498003 0.492612 0.452276 0.441494 0.420727 0.417931 0.408946 0.371605 0.345248 0.294329
*MYO6*	single nucleotide variant (701) deletion (8) insertion (3) indel (104)	missense variant (6) synonymous variant (3) intron variant (715) nc transcript variant (29) 5 prime UTR variant (1) 3 prime UTR variant (28) 500 B downstream variant (2) 2 KB upstream variant (10)	uncertain-significance (2) likely-benign (15) benign (14)	/	rs276696 rs9360941 rs2748949 rs2842550 rs2647404 rs9360958 rs7742137 rs2842554 rs6920348 rs6903077	intron variant intron variant not specified 2 KB upstream variant intron variant intron variant nc transcript variant, 3 prime UTR variant not specified intron variant intron variant	C,T A,G,T C,A,G G,A,T G,A A,C,G C,A,T C,A,T T,A,G A,C,G	C A G A A A C C G G	0.48103 0.476238 0.471645 0.470847 0.470847 0.465455 0.449481 0.422324 0.421526 0.421526
*NCR3*	single nucleotide variant (27) deletion (1) indel (3)	missense variant (2) synonymous variant (4) intron variant (17) nc transcript variant (2) 5 prime UTR variant (2) 3 prime UTR variant (5) 500 B downstream variant (5) 2 KB upstream variant (4)	risk-factor (1)	/	rs1052248 rs2736191 rs2736190 rs3087617 rs986475 rs11575842 rs3896375 rs41268892 rs11575836 rs41268888	nc transcript variant, 3 prime UTR variant, 500 B downstream variant 2 KB upstream variant 2 KB upstream variant nc transcript variant, 3 prime UTR variant, 500 B downstream variant 3 prime UTR variant, 500 B downstream variant intron variant intron variant intron variant 5 prime UTR variant intron variant	T,A,C C,G T,A,C,G A,T A,G,T G,A G,A G,A,C A,G G,C	A G T T G A A A G C	0.301717 0.228235 0.174121 0.105232 0.104832 0.0782748 0.0778754 0.0778754 0.0776757 0.0760783
*PPARG*	single nucleotide variant (633) deletion (8) insertion (1) indel (65)	missense variant (1) synonymous variant (1) intron variant (556) 5 prime UTR variant (4) 3 prime UTR variant (27) 500 B downstream variant (2) 2 KB upstream variant (6)	likely-benign (2)	7	rs147070788 rs7618026 rs7618046 rs17819328 rs1152003 rs4684104 rs10602803 rs4684854 rs2960420 rs2959269	not specified not specified not specified not specified not specified not specified not specified not specified not specified intron variant	G,A,T T,C T,A,C T,A,G G,C A,C,G,T G,A,T G,A,C,T C,G T,A,C	A C C G C A A C G C	0.495407 0.491414 0.491014 0.489816 0.480232 0.478035 0.460863 0.458866 0.457867 0.455471
*SARM1*	single nucleotide variant (70) deletion (1) indel (10)	intron variant (64) 5 prime UTR variant (1) 3 prime UTR variant (17) 500 B downstream variant (1) 2 KB upstream variant (9)	uncertain-significance (1) benign (11)	/	rs2027993 rs967645 rs2239911 rs2239907 rs7212349 rs7212510 rs2239908 rs4795434 rs4795433 rs4794828	intron variant intron variant 3 prime UTR variant 3 prime UTR variant 2 KB upstream variant intron variant 3 prime UTR variant intron variant intron variant intron variant	G,C,T C,A,G,T G,C,T T,A,C,G T,A,C T,A G,A,C,T G,T C,A,G,T G,A,T	G C G T T T G G C G	0.469449 0.46845 0.466054 0.451478 0.423522 0.405351 0.394768 0.394369 0.389776 0.389177
*TCF20*	single nucleotide variant (496) deletion (6) insertion (4) indel (67)	missense variant (5) synonymous variant (5) intron variant (438) nc transcript variant (7) splice donor variant (1) 5 prime UTR variant (4) 3 prime UTR variant (2) 500 B downstream variant (7) 2 KB upstream variant (36)	benign (4)	/	rs134885 rs134886 rs760648 rs134867 rs134899 rs134891 rs134889 rs134888 rs6002655 rs86669	intron variant intron variant intron variant intron variant not specified intron variant intron variant intron variant intron variant 2 KB upstream variant	C,A,G,T A,C,G,T G,A,C,T T,A,C T,A,G T,C A,C,G,T C,G,T C,A,G,T C,G,T	C A A T T T A C C T	0.47484 0.474641 0.467252 0.462061 0.460264 0.454273 0.453275 0.448083 0.441693 0.439497
*TCF7L2*	single nucleotide variant (846) deletion (11) insertion (9) indel (129)	missense variant (4) frameshift variant (1) synonymous variant (1) intron variant (844) 5 prime UTR variant (6) 3 prime UTR variant (8) 500 B downstream variant (5) 2 KB upstream variant (15)	drug-response (1) risk-factor (2) benign (2)	14	rs720785 rs7918976 rs11196171 rs11196170 rs2296784 rs720784 rs7897438 rs290476 rs10885399 rs61875109	intron variant not specified intron variant intron variant intron variant intron variant intron variant intron variant intron variant intron variant	G,A,C C,A,G,T A,C,G G,A,C,T T,C A,C,G,T C,A,G,T G,A,C,T T,A,G C,A,G,T	G A A G T A A T A A	0.499401 0.498602 0.495607 0.494609 0.494409 0.49401 0.478235 0.477835 0.477436 0.477236
*TNF*	single nucleotide variant (9) indel (4)	synonymous variant (1) intron variant (7) 3 prime UTR variant (1) 500 B downstream variant (1) 2 KB upstream variant (3)	benign (1)	39	rs1800610 rs3093662 rs3093664 rs361525 rs3093661 rs673 rs3093665 rs2228088 rs41297589	intron variant intron variant intron variant 2 KB upstream variant intron variant 2 KB upstream variant 3 prime UTR variant synonymous variant 2 KB upstream variant	G,A A,G A,G G,A G,A,C G,A A,C G,A,C,T T,A	A G G A A A C T A	0.100439 0.0798722 0.0788738 0.0609026 0.0521166 0.0191693 0.01877 0.0175719 0.0105831

Variant types, molecular consequences, most severe clinical significances, and top 10 genetic variants with highest frequencies were identified in the Variation Viewer database. The number of variants that have already been shown to have specific pharmacologic relevance was retrieved from the PharmGKB database.

**Table 4 nutrients-16-01362-t004:** GWAS-reported variants of selected top-priority candidate genes (DEGs, URs, or both) and their frequencies in the global population.

Variant and Risk Allele	Mapped Gene/s in GWAS	Gene: Consequence in dbSNP	Global Frequency in 1000 Genomes	Associated Trait in GWAS	SNP Identified in GWAS	Gene Identified in Flavonoid Study/-ies	Number of Citations in dbSNP
rs671; A	*ALDH2*	*ALDH2*: Missense Variant	A = 0.0357	Hypertension	GCST011141	*ALDH2*: DEG in Paper 1	293
rs445925; G rs445925; A	*APOE* (also *APOC1*) *APOE* (also *APOC1*)	*APOC1*: 2KB Upstream Variant *APOC1*: 2KB Upstream Variant	G = 0.8502 A = 0.1498	Atherosclerosis Atherosclerosis	GCST001231 GCST001231	*APOE*: UR in Papers 1, 4, and 5	28
rs10745332; A rs17030613; A	*CAPZA1* *CAPZA1*	*CAPZA1*: Intron Variant *CAPZA1*: Intron Variant	A = 0.8131 A = 0.7678	Hypertension Hypertension	GCST002627 GCST007707	*CAPZA1*: DEG in Papers 2, 3, and 4	0 2
rs62524579; A rs12679242; T rs6418; A rs1799998; G	*CYP11B2* (also *LY6E-DT*) *CYP11B2* *CYP11B2* (also *GML*) *CYP11B2* (also *LY6E-DT*)	None *CYP11B2*: Intron Variant *CYP11B2*: Intron Variant *CYP11B2*: 2KB Upstream Variant	A = 0.4794 T = 0.3470 A = 0.6450 G = 0.3472	Hypertension Hypertension Hypertension Hypertension	GCST007707 GCST007707 GCST007707 GCST011141	*CYP11B2*: UR in Paper 4	1 0 0 35
rs12941507; C	*GNA13* (also *AMZ2P1*)	None	C = 0.0647	Hypertension	GCST011952; GCST011953	*GNA13*: DEG in Papers 2 and 3	0
rs4722172; G	*IL6* (also *MTCYBP42*)	None	G = 0.0595	Atherosclerosis	GCST008474; GCST90061371	*IL6*: UR in Papers 4 and 5	1
rs4728142; A	*IRF5* (also *KCP*)	None	A = 0.2945	Hypertension	GCST006023	*IRF5*: DEG in Paper 5; UR in Papers 4 and 5	54
rs6511720; T rs138294113; C	*LDLR* *LDLR* (also *SMARCA4*)	*LDLR*: Intron Variant; *LDLR-AS1*: 2KB Upstream Variant None	T = 0.0917 C = 0.9095	Atherosclerosis Atherosclerosis	GCST001231 GCST008474; GCST90061371	*LDLR*: DEG in Paper 1; UR in Papers 2 and 5	73 0
rs322; A	*LPL*	*LPL*: Intron Variant	A = 0.7079	Atherosclerosis	GCST008474; GCST90061371	*LPL*: DEG in Paprer 2	0
rs1973765; T rs569550; T rs661348; T rs4980389; A	*LSP1* *LSP1* *LSP1* *LSP1*	*LSP1*: Intron Variant *LSP1*: Intron Variant * *LSP1*: Intron Variant *LSP1*: Intron Variant **	T = 0.5641 T = 0.5765 T = 0.6182 A = 0.4267	Hypertension Hypertension Hypertension Hypertension	GCST007707 GCST007707 GCST007707 GCST007707	*LSP1*: DEG in Papers 3 and 4; UR in Paper 5	0 1 7 0
rs139537100; C	*MKNK1* (also *MOB3C*)	*MOB3C*: Intron Variant *** Allele Frequency Aggregator	C = 0.999938	Hypertension	GCST010477	*MKNK1*: DEG in Paper 4; UR in Paper 4	0
rs566125; T	*MMP3*	*MMP3*: Intron Variant	T = 0.0755	Atherosclerosis	GCST008474; GCST90061371	*MMP3*: UR in Paper 4	2
rs17367504; not reported	*MTHFR*	*MTHFR*: Intron Variant	/	Hypertension	GCST009685	*MTHFR*: DEG in Paper 5	33
rs3798440; A x rs9350602; C	no mapped genes x *MYO6* (SNP x SNP interaction)	rs3798440; *MYO6*: Intron Variant **** rs9350602; *MYO6*: Intron Variant ****	rs3798440; A = not present rs9350602; C = 0.8972	Hypertension	GCST001085	*MYO6*: DEG in Paper 4; UR in Paper 5	0 0
rs2515920; T	*NCR3* (also *UQCRHP1*)	*NCR3*: 2KB Upstream Variant	T = 0.0495	Hypertension	GCST010477	*NCR3*: DEG in Papers 2 and 5; UR in Paper 5	0
rs17036160; C	*PPARG*	*PPARG*: Intron Variant *****	C = 0.9319	Arterial stiffness	GCST008403	*PPARG*: UR in Papers 4 and 5	3
rs704; A	*SARM1* (also *VTN*)	*VTN*: Missense Variant	A = 0.5551	Hypertension	GCST90000064	*SARM1*: DEG in Papers 4 and 5	9
rs17478227; not reported	*TCF20*	*TCF20*: Intron Variant	/	Arterial stiffness	GCST007846	*TCF20*: DEG in Papers 1 and 4; UR in Paper 4	1
rs7903146; T	*TCF7L2*	*TCF7L2*: Intron Variant ******	T = 0.2278	Atherosclerosis	GCST008474; GCST90061371	*TCF7L2*: DEG in Paper 1; UR in Papers 3 and 4	660
rs769177; G	*TNF* (also *LTB*)	None	rs769177; G = not present	Hypertension	GCST010477	*TNF*: UR in Papers 2, 4 and 5	7

* genic_upstream_transcript_variant, intron_variant; ** intron_variant, 5_prime_UTR_variant; *** 5_prime_UTR_variant, intron_variant; **** intron_variant, genic_downstream_transcript_variant; ***** genic_upstream_transcript_variant, intron_variant, upstream_transcrip; ****** intron_variantgenic_upstream_transcript_variant.

## Data Availability

The original contributions presented in the study are included in the article/Appendix A, further inquiries can be directed to the corresponding author.

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
