# Peer review of "Integrated Analysis of Genomic and Genome-Wide Association Studies Identified Candidate Genes for Nutrigenetic Studies in Flavonoids and Vascular Health: Path to Precision Nutrition for (Poly)phenols"

_nutrients, 2024, doi:10.3390/nu16091362_

Round 1
Reviewer 1 Report
Comments and Suggestions for Authors
Ruskovska and coll presented the research: “Integrated analysis of genomic and genome-wide association studies identified candidate genes for nutrigenetic studies in flavonoids and vascular health: path to precision nutrition for (poly) phenols”, aimed to identify possible genes (and their polymorphisms) associated with interindividual variability in response to flavonoids intake.
This study faces a very interesting topic by extracting results from an integrative analysis of transcriptomic data with GWAS, so that Authors identify several candidate genes.
Although the topic is interesting, the research presents some weaknesses.
Unfortunately, this article is rather hard to understand, often confusing for the reader. In addition, the figures are not helpful for the comprehension of the text.
Moreover, the Authors identify and select only five human intervention studies (too few) reporting positive effects of flavonoids on vascular function together with global genomic changes analyzed using microarray techniques.
Please explain what this study adds to the current literature.
The language should be revised.
Comments on the Quality of English LanguageThe language should be revised.
Author Response
Reviewer 1
Ruskovska and coll. presented the research: “Integrated analysis of genomic and genome-wide association studies identified candidate genes for nutrigenetic studies in flavonoids and vascular health: path to precision nutrition for (poly) phenols”, aimed to identify possible genes (and their polymorphisms) associated with interindividual variability in response to flavonoids intake.
-This study faces a very interesting topic by extracting results from an integrative analysis of transcriptomic data with GWAS, so that Authors identify several candidate genes.
We thank the Reviewer for this very positive remark.
-Although the topic is interesting, the research presents some weaknesses.
Unfortunately, this article is rather hard to understand, often confusing for the reader. In addition, the figures are not helpful for the comprehension of the text.
Following the Reviewer’s comments, we modified the manuscript to make the reading and understanding clearer and more comprehensive. In addition, we have included explanations in figure legends to make the figures more comprehensive.
-Moreover, the authors identify and select only five human intervention studies (too few) reporting positive effects of flavonoids on vascular function together with global genomic changes analyzed using microarray techniques.
We do agree with the Reviewer that this number of the papers is low, however that is the number of studies which performed both genomic analyses and evaluated cardiovascular health effect of flavonoids. We thank the Reviewer for pointing out this issue, which we have addressed in the limitation paragraph.
-Please explain what this study adds to the current literature.
Studies that have been performed so far in the field of (poly)phenols aimed to identify health properties of these bioactives, and/or ascertain genomic changes following intake of (poly)phenols to decipher the molecular mechanisms of action. Moreover, few studies also showed that there is variability in health properties of (poly)phenol intake depending on different factors, including genetic polymorphism. Only a handful of studies showed associations between variably in response to (poly)phenol intake and genetic polymorphism. Our study is first that combined GWAS and transcriptomic data in the field of (poly)phenols and identified candidate genes for future nutrigenetic studies aiming to assess interindividual variability of the effects of flavonoids on vascular health and genetic polymorphism. Therefore, the data analyses of this paper provide foundation for precision nutrition research in the field of (poly)phenols.
The language should be revised.
We revised the manuscript for language.
Reviewer 2 Report
Comments and Suggestions for Authors
See attached document.

Author Response
Reviewer 2
The manuscript reports a complex, multilevel analysis that incorporates multiple data sources from published literature, GWAS, and omics databases. The study is well-written, its methodology seems appropriate, and reports some interesting findings that may help design sophisticated and focused future studies to characterize the complex interplay between genetic factors and dietary flavonoids on vascular health.
We thank the Reviewer for their very positive comments.
Few points need to be addressed, however.
- It is not clear why upstream regulators (URs) where included in the analysis—unless the researchers hypothesize that URs may have a direct effect on vascular physiology (other than through DEGs). This point should be explained.
We thank the Reviewer for their pertinent comment. Indeed, upstream regulators (URs) are molecules capable of regulating the expression of differentially expressed genes (DEGs). Among these URs, identified using the QIAGEN Ingenuity Pathway Analysis (IPA) software, there are proteins that are inherently prone to genetic variability, potentially serving as a source for significant interindividual variabilities in flavonoids intake and vascular health effects. We have incorporated this explanation into the discussion section of the revised manuscript.
- Related to (1) how many of the final list of 20 candidate genes that were identified by the authors belong to DEGs, URs?
We do agree with the Reviewer that this is important information, which we have included in the penultimate column of Table 4.
- Also related to (1), does the URs include epigenetic factors (e.g., methylation level)?
We do agree with the Reviewer that modulations in DNA methylation profile present an important potential mechanism underlying interindividual variabilities in flavonoids intake and vascular health effects. However, in this study we did not address modulations in DNA methylation, as stated in the limitation paragraph at the end of the revised manuscript.
- The authors interrogated GWAS catalog for “genes that are associated with vascular dysfunction” (line 143). Wouldn’t looking for genes that are associated with “vascular function” instead cast as wider net? Is it possible that flavonoids promote vascular health by weakening the effect of the “bad genes” and/or promoting the effect of the “good genes”.
A genome-wide association study (GWAS) involves genome-wide analysis of genotypes of a large number of individuals to identify variants associated with a specific disease or health-related trait/dysfunctions compared to healthy individuals. As such, searching of GWAS catalogue allows interrogation of different pathologies only, including the traits associated to vascular dysfunction (hypertension, atherosclerosis, and arterial stiffness), which we analyzed in this study. We agree with the Reviewer that studying good genes for vascular function can give a valuable information on health promoting effects of (poly)phenols. We tried to address this point in one of our previous studies “Systematic analysis of nutrigenomic effects of polyphenols related to cardiometabolic health in humans – Evidence from untargeted mRNA and miRNA studies”, (https://doi.org/10.1016/j.arr.2022.101649; results presented in Figure 11).
- For Table 1, it will be helpful to add some more information to description of the study sample. For example, the participants in studies 3 and 4 include smokers or not? If not, it will be interesting to examine whether the pathways to which the 72 genes that were common between studies 3 and 4 (healthy), are different from the ones to which the 45 genes common between studies 1 and 2 (overweight/smoking).
We thank the Reviewer for pointing out this important issue. As recommended, we added a column describing the study populations.
Minor:
The numbering of the pages is incorrect.
We apologize for this inconvenience, which we believe is a result of introducing a page with landscape orientation. We corrected this issue in the revised version of the manuscript.
Reviewer 3 Report
Comments and Suggestions for Authors
Dear Authors,
the work was well conducted and a lot of relevant data was presented. It was presented data about genes that potentially could have some capacity to underlie the interindividual variability of the vascular effects in response to flavonoids intake as candidate genes for future nutrigenetic studies on flavonoids and vascular health. However, as natural product researcher I missed some data correlation to the many different chemical classes of flavonoids, including the variation of bioavailability of them due to their polarity differences, since some of them can be mono or biglycosylated. the term flavonoids can include a wide variety of compounds and substances with differences in absorption and biological effect. Thus, it would be very important to insert some information about this in the text.
- In my opinion the introduction could be reduced, and include some information about the different types of flavonoids and their absorption/bioavailability correlation as nutrients and theirs expected biological effects. (Also in the discussion section).
Author Response
Reviewer 3
The work was well conducted and a lot of relevant data was presented. It was presented data about genes that potentially could have some capacity to underlie the interindividual variability of the vascular effects in response to flavonoids intake as candidate genes for future nutrigenetic studies on flavonoids and vascular health.
We thank the Reviewer for their very positive comment.
However, as natural product researcher I missed some data correlation to the many different chemical classes of flavonoids, including the variation of bioavailability of them due to their polarity differences, since some of them can be mono or biglycosylated. The term flavonoids can include a wide variety of compounds and substances with differences in absorption and biological effect. Thus, it would be very important to insert some information about this in the text.
As proposed by the Reviewer, we added several sentences, which are placed in the first paragraph of the revised manuscript, as well as two additional references (ref. no. 4 and ref. no. 6 in the revised manuscript), to address the differences in chemical structures among (poly)phenols and their bioavailability.
- In my opinion the introduction could be reduced and include some information about the different types of flavonoids and their absorption/bioavailability correlation as nutrients and theirs expected biological effects (also in the discussion section).
We have reduced the introduction as kindly suggested.
Reviewer 4 Report
Comments and Suggestions for Authors
- The manuscript titled "Integrated analysis of genomic and genome-wide association studies identified candidate genes for nutrigenetic studies in flavonoids and vascular health: path to precision nutrition for (poly)phenols" presents an original contribution to the field of nutritional genomics. The methodology employed in the study is appropriate to address the research aims and stands out for its innovative approach of integrating information from previous studies to provide practical insights.
Minor Improvements:
- Figures and tables should be self-explanatory and include all necessary information without relying heavily on the main text for context.
- - Please consider including the significance for all acronyms.
- - In Figure 2, provide clear identification for sets (e.g., set 1, set 2, set 3...).
- - Similarly, in Figure 3, improve the identification or proper reference for Paper 1, Paper 2, Paper 3....
- - In Figures 2 and 4, specify the software or website used to draw.
- These enhancements will improve the overall readability and comprehensibility of the figures and tables, ensuring that readers can grasp the key findings more effectively.
Author Response
Reviewer 4
The manuscript titled "Integrated analysis of genomic and genome-wide association studies identified candidate genes for nutrigenetic studies in flavonoids and vascular health: path to precision nutrition for (poly)phenols" presents an original contribution to the field of nutritional genomics. The methodology employed in the study is appropriate to address the research aims and stands out for its innovative approach of integrating information from previous studies to provide practical insights.
We thank the Reviewer for their very positive comments.
Additionally, we have carefully considered all suggestions for minor improvements to the manuscript, which we greatly appreciate.
Minor Improvements:
- Figures and tables should be self-explanatory and include all necessary information without relying heavily on the main text for context.
We thank the Reviewer for this comment, following which we have improved explanations of tables and figures.
- Please consider including the significance for all acronyms.
On the first page, just below the “keywords”, we have included the list of acronyms. Also, in the section Material and Methods we have included the following statements:
- “Gene names and symbols were searched in GeneCards database… (followed by the link to the website and a reference).”
- “The names of canonical pathways are presented as they appear in interrogated databases.”
- In Figure 2, provide clear identification for sets (e.g., set 1, set 2, set 3...).
We have clearly identified the gene sets.
- Similarly, in Figure 3, improve the identification or proper reference for Paper 1, Paper 2, Paper 3....
We have improved identification of Paper 1-5.
- In Figures 2 and 4, specify the software or website used to draw.
We have specified the website used.
These enhancements will improve the overall readability and comprehensibility of the figures and tables, ensuring that readers can grasp the key findings more effectively.